# MaxCode: A Max-Reward Reinforcement Learning Framework for Automated Code Optimization

## Abstract

Large Language Models (LLMs) demonstrate strong capabilities in general coding tasks but encounter two key challenges when optimizing code: (i) the complexity of writing optimized code (such as performant CUDA kernels and competition-level CPU code) requires expertise in systems, algorithms and specific languages and (ii) requires interpretation of performance metrics like timing and device utilization beyond binary correctness. In this work we explore inference-time search algorithms that guide the LLM to discover better solutions through iterative refinement based on execution feedback. Our approach called **MaxCode** unifies existing search methods under a max-reward reinforcement learning framework, making the observation and action-value functions modular for modification. To enhance the observation space, we integrate a natural language critique model that converts raw execution feedback into diagnostic insights about errors and performance bottlenecks, and the best-discounted reward seen so far. Together, these provide richer input to the code proposal function. To improve exploration during search, we train a generative reward-to-go model using action values from rollouts to rerank potential solutions. Testing on the KernelBench (CUDA) and PIE (C++) optimization benchmarks shows that **MaxCode** improves optimized code performance compared to baselines, achieving 20.3% and 10.1% relative improvements in absolute speedup value and relative speedup ranking, respectively.

## 1 Introduction

Recent advancements in Large Language Models (LLMs) have revolutionized automatic code generation, driving the development of specialized coding tools such as Claude Code (Cla, b), Qwen3-Coder (Yang et al., 2025), and Code Llama (Rozière et al., 2023). The verifiable nature of code through execution testing has enabled researchers to leverage execution feedback for improving LLM-based code generation systems. This approach has proven particularly valuable for **code optimization** (Ouyang et al., 2025; Madaan et al., 2023), where LLM-based optimization methods must satisfy dual objectives: ensuring correctness while maximizing *performance* metrics such as execution time and resource utilization. The practical impact of code optimization extends far beyond academic benchmarks—optimizing CUDA kernels for fundamental operations can yield substantial computational savings, potentially reducing GPU hours by orders of magnitude when deployed at scale Dao (2023); Shah et al. (2024); Wang et al. (2024).

Code optimization presents two fundamental challenges that distinguish it from general coding tasks: 1) the intrinsic complexity of generating optimized code demands sophisticated reasoning about algorithmic trade-offs, memory access patterns, and hardware-specific optimizations that make it more difficult for LLMs to produce correct solutions, and 2) the need to interpret multifaceted performance feedback (timing, hardware utilization, and resource consumption metrics) beyond binary compilation and execution correctness. For example, Figure 1 shows two code samples generated by `Deepseek-R1` (DeepSeek-AI et al., 2025) that optimize a CUDA kernel implementing a chain of PyTorch operators using drastically different approaches. The left sample fuses operator subsets sequentially before chaining sub-kernels, while the right sample fuses all operators simultaneously—yet both achieve nearly identical wall-clock performance, illustrating the non-obvious relationship between implementation strategy and performance outcomes that complicates

Figure 1: Example optimization code generated by `DeepSeek-R1` on a KernelBench problem

optimization decisions. This demonstrates that viable optimization solutions exhibit high diversity in structure and semantics, requiring deep understanding of the problem domain, programming language semantics, and underlying hardware architecture. Moreover, raw performance feedback provides insufficient diagnostic information: knowing that code runs 20% slower offers no insight into specific bottlenecks (memory bandwidth, compute utilization, or algorithmic inefficiency) or actionable remediation strategies. Consequently, even state-of-the-art LLMs with advanced coding capabilities struggle significantly with kernel optimization tasks (Ouyang et al., 2025).

To address these challenges, we first cast the problem of performance improving code optimization as *max-reward reinforcement learning*, which captures the notion of attaining best performance as opposed to cumulatively rewarded performance Veviurko et al. (2024). This formulation warrants inclusion of best-discounted reward in the observation space, for both learning and inference. Inspired by recent work on LLM self-refinement through natural language critique (Xie et al., 2025), we enrich the observation space with feedback from a critique model that analyzes optimized code and raw execution feedback to generate diagnostic insights and actionable refinement suggestions (Figure 2). We then define a *max-reward inference operator* to perform inference with a fixed policy, and instantiate the inference operator using multiple search algorithms to guide off-the-shelf LLMs in exploring and iteratively refining solutions. We call our approach **MaxCode** - a formulation that combines critique-augmented observation space with best-discounted reward to guide inference time search, enabling more effective exploration of the optimization solution space.

In code optimization, the evaluation of generated solutions demands computational resources and often becomes the limiting factor to effectively scale the search under a given computation and time budget. So, we additionally explore use of a trained generative Value/Reward-to-go model (Mahan et al., 2024) which predicts the V-value of any search trajectory prefix, i.e., the expected maximum future performance on that search branch given a proposed action (code revision). We train the reward-to-go model with roll-outs sampled from our tree searches. The learned reward model can be integrated at each search step by oversampling candidate refinements, filtering with predicted reward, and retaining only the most promising samples for evaluation and continuation. As a result, we enable search process to effectively explore more candidates under a certain evaluation budget.

We evaluate **MaxCode** formulation on two code optimization tasks: kernel code optimization (Ouyang et al., 2025) and competitive C++ code optimization (Madaan et al., 2023). With extensive experiments, we demonstrate that by integrating with our proposed max-reward RL formulation, the performance of existing methods can be significantly boosted. In particular, combining the best-performing search method (CUDA LLM (Chen et al., 2025)) with **MaxCode** yields relative speedup improvements of 27.3%, 11.0% and 22.5% on KernelBench level 1, level 2 and PIE, respectively.

In summary, our main contributions are as follows:

- We formalize code optimization as a max-reward reinforcement learning problem and augment the observation space with two key components: (i) the best-discounted reward seen so far, and (ii) a natural language critique generated by a dedicated critique model that provides performance diagnosis and optimization suggestions based on code analysis and execution results. This enables more targeted search and provides actionable optimization suggestions based on code analysis and execution results.

- We define a max-reward inference operator and implement it through various search algorithms for fixed-policy inference. Our framework leverages the augmented observation space to enable effective exploration of the optimization solution space. Through empirical evaluation on kernel code optimization and competitive C++ optimization tasks, we demonstrate significant performance improvements over existing methods when integrating with our proposed framework.

- We propose a categorical Value/Reward-to-go model that predicts the expected maximum future performance of search trajectories, enabling efficient candidate filtering and resource optimization through informed trajectory selection.

## 2 MAXCODE: MAX-REWARD RL FRAMEWORK FOR CODE OPTIMIZATION

**The Markov Decision Process** We formulate the performance improving code optimization process as a Markov Decision Process (MDP) with an expanded state space that incorporates initial code, current code, execution feedback, and language model critiques. We illustrate in Figure 2 the MDP process with max-reward formulation. Formally, we define our MDP with the tuple $(\mathcal{S}, \mathcal{A}, P, R, \gamma, \rho_0)$, where the state space $\mathcal{S}$ is defined as the product space $\mathcal{X}_0 \times \mathcal{X} \times \mathcal{E} \times \mathcal{C}$, where $\mathcal{X}_0$ represents the problem description along with the initial code state, $\mathcal{X}$ represents the space of possible current code states, $\mathcal{E}$ represents execution feedback, and $\mathcal{C}$ represents language model critiques Each state $s_t = (x_0, x_{t-1}, e_{t-1}, c_{t-1})$ provides a representation of the initial code, current code, its execution results, and associated natural language critique. The initial state distribution $\rho_0 : X_0 \to [0, 1]$ defines the probability distribution over starting code states, which is assumed uniform over the code samples present in the considered benchmarks, and $\gamma \in (0, 1]$ is the discount factor, which we set to 1 because of finite horizon rollouts. The reward function $R : \mathcal{S} \times \mathcal{A} \times \mathcal{S} \to \mathbb{R}$ defined as $R(s_t, a_t, s_{t+1}) = f(e_{t+1})$, where $f$ evaluates code performance based on execution feedback $e_{t+1}$, returning higher values for improved performance.

The action space $\mathcal{A} = \mathcal{X}$ corresponds to the space of possible code modifications. Unlike standard RL, the policy $\pi_\theta$ is a large language model (LLM) with frozen parameters $\theta$ that operates autoregressively on states. Given $s_t$, the policy implicitly applies a sequence of token-level edits and produces a distribution over complete code candidates: $\pi_\theta(x \mid s_t) : \mathcal{X} \to \Delta(\mathcal{X})$. Due to stochastic token sampling, the same state $s_t$ may yield multiple distinct candidates $\{x_{t+1}^{(1)}, \ldots, x_{t+1}^{(M)}\}$.

The transition function $P$ captures two sources of stochasticity: policy stochasticity from autoregressive token sampling in $\pi_\theta$, and the environment stochasticity from $\pi_\theta$ - the LLM-based critique generator. Formally, after the policy outputs $x_{t+1}$, the environment produces the next state by augmenting the trajectory with execution results and critique: $P(s_{t+1} \mid s_t, x_{t+1}) = P(e_{t+1}, c_{t+1} \mid x_{t+1}), \delta[s_{t+1} = (x_0, x_{t+1}, e_{t+1}, c_{t+1})]$, where $e_{t+1}$ is obtained from running $x_{t+1}$ on the target hardware, $c_{t+1}$ is generated by a separate LLM that produces a natural-language critique conditioned on $(x_{t+1}, e_{t+1})$, and $\delta[\cdot]$ enforces deterministic update of the state components.

Following the max-reward RL formulation Veviurko et al. (2024), we define the return from time $t$ as $\hat{G}_t = \max k \geq 1 \gamma^{k-1} r_{t+k}$, which captures the best performance eventually achieved from time $t$. With $u \in \mathbb{R}$ as an auxiliary real variable representing the best discounted reward obtained so far, max-reward value functions under policy $\pi$ are given by

$$V^\pi(s, u) = \mathbb{E}_\pi \left[ \max(u, \hat{G}_t) \mid s_t = s \right], \tag{1}$$

$$Q^\pi(s, a, u) = \mathbb{E}_\pi \left[ \max(u, \hat{G}_t) \mid s_t = s, a_t = a \right]. \tag{2}$$

**Remark** In max-reward RL, the optimal policy maximizing expected return from the initial state should depend not only on the current state, but also on the rewards obtained so far. The auxiliary

variable $u \in \mathbb{R}_{\geq 0}$ representing the best discounted reward achieved so far is crucial for maintaining the Markov property under the max-reward objective.

**Execution and Critique** In our setup, the Executor $EX$ evaluates input $x$ against test cases $Y$, producing feedback $e = \{e_{1a}, e_{1b}, e_{2a}, e_{2b}\}$, where $e_{1a}$ indicates binary correctness, $e_{1b}$ provides contrastive correctness details (*e.g.* difference in the output of the current and previous code for some test cases), $e_{2a}$ is a numerical performance indicator (e.g., running time and / or relative speed-up against the previous code, and $e_{2b}$ contains contrastive performance details. For standard coding tasks, feedback is limited to correctness components ($e = e_{1a}, e_{1b}$) and serves purely as an evaluation metric. In optimization tasks, where initial solutions are typically correct, the focus shifts to performance metrics $e_2$, as the initial solution remains a fallback option if optimization fails. Given that raw execution feedback can often prove to be less informative (Xie et al., 2025), we introduce a critic model ($\pi_c$) to generate natural language critiques that provide both diagnostic insights into potential bugs and performance bottlenecks, as well as actionable refinement suggestions.

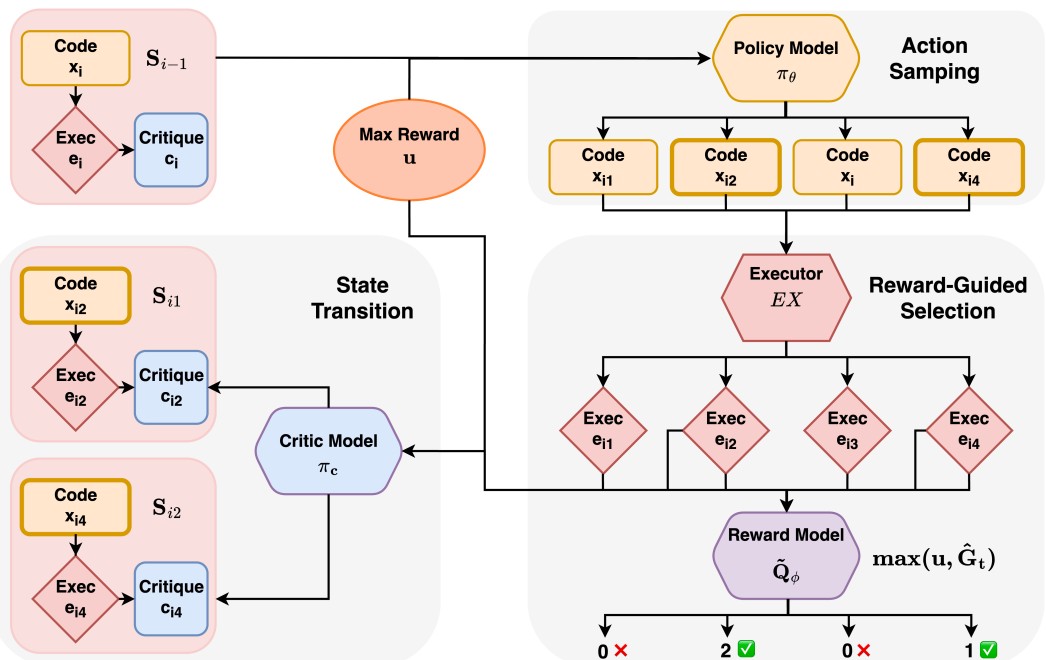

Figure 2: Illustration of the MaxCode search method

## 2.1 MAX-REWARD INFERENCE OPERATOR

One can obtain an optimized code with a budget $K$ by sampling trajectories $\{\tau_1, \ldots, \tau_K\}$ and selecting:

$$\tau^* = \arg\max_{\tau_i} \hat{G}(\tau_i) = \arg\max_{\tau_i} \max_{t \in [1,T]} \gamma^{t-1} f_r(e_t^{(i)}) \tag{3}$$

Here, $\tau_i = (s_0, s_{i_1}, \ldots, s_{i_{T-1}}, s_T)$ represents a trajectory run for T time steps. Instead, we propose searching for the best optimized code under an extended MDP with state space $(s, u)$. To do so, we define a *max-reward inference operator* $\mathcal{T}^*$ applied to a fixed policy $\pi_\theta$ as:

$$\mathcal{T}^*(\pi_\theta)(s, u) \approx \arg\max_a Q^{\pi_\theta}(s, a, u) \tag{4}$$

where $Q^{\pi_\theta}(s, a, u) = \mathbb{E}_{\pi_\theta}[\max(u, \hat{G}_t) \mid s_t = s, a_t = a]$ is the max-reward Q-function with auxiliary variable $u$ representing the best discounted reward achieved so far. The operator performs one step of greedy policy improvement in an extended MDP with state space $(s, u)$. Unlike standard

policy improvement that operates on states $s$ alone, our operator considers both the current state and the reward history encoded in $u$, enabling decisions that depend on the quality of solutions found so far.

## 2.2 Max-Reward Search

We now show how to approximately implement $\mathcal{T}^*$ with inference time search by adopting and repurposing various prior work under the proposed max-reward RL formulation. Given a code optimization problem with test cases $Y$, generator LLM $\pi_\theta$, and critic LLM $\pi_c$, we define initial state $s_0 = (x_0, \emptyset, \emptyset, \emptyset) \in \mathcal{X}_0 \times \mathcal{X} \times \mathcal{E} \times \mathcal{C}$, where $x_0$ is the problem statement with the initial code, and search states as $s_i = (x_0, x_i, e_i, c_i) \in \mathcal{S}$ represent the current optimization candidate with its execution feedback and critique. For each state $s_i$, we maintain $u_i = \max_{j \leq i} \gamma^{d_j} f(e_j)$ tracking the best discounted reward achieved along the trajectory to $s_i$, where $d_j$ is the depth of state $j$. With this setting, we reformulate the following methods for max-reward search:

**Effi-Learner** Given $s_0$ (Huang et al., 2024) proposed to 1) first sample an initial action $x_1$ from $\pi_\theta(s_0)$ and obtain its execution feedback $e_1 = EX(x_1, Y)$; 2) generate a refinement action $x_2$ from $\pi_\theta(x_0, x_1, e_1)$ as the final solution.

*Max-Reward Reformulation*: Under Max-Reward formulation, we reformulate Effi-learner to 1) additionally obtain the critique $c_1 \sim \pi_c(x_1, e_1)$ and form the successor state $s_1 = (x_0, x_1, e_1, c_1)$ via transition function $P(s_1 \mid s_0, x_0)$; and 2) generate the final solution $x_2$ from $\pi_\theta(s_1)$. Noting that we are not adding the we leverage the maintained best discounted reward $u_i = \max_{j \leq i} \gamma^{d_j} f(e_j)$ since Effi-Learner performs only 2 rounds of optimization thus $u_i$ is already encoded in $e_1$.

**CUDA-LLM** Chen et al. (2025) proposes a beam-search based method that given a state $s_i' = (x_0, x_i, e_i)$, sampling $k$ candidate actions $x_{i1} \ldots x_{ik}$, obtaining execution feedback $e_{i_j} = EX(x_{i_j}, Y)$ and 1) if any of the candidates is correct, i.e. the speedup $f(e_{ij}) > 1$, select the best-performing candidate to proceed with, i.e. $x_m$ with $m = \arg\max_x(f(e_x))$, and form the new state $s_{i+1}' = (x_0, x_m, e_m)$; 2) if none of the candidates are correct, formulate intermediate states $s_{ij}' = (x_0, x_{ij}, e_{ij})$ and iteratively refine them by sampling and executing (in parallel) single refinement candidates for each intermediate state until at least one of the refinement is correct. Then it discard all intermediate states and obtain $s_{i+1}$ as in 1).

*Max-Reward Reformulation*: Under Max-Reward formulation, we 1) enhance each state $s_i' = (x_0, x_i, e_i)$ with natural language critiques obtained by $\pi_c$ to obtain complete states $s_i = (x_0, x_i, e_i, c_i)$; 2) when sampling the next action at each state $s_i$, we leverage the maintained best discounted reward $u_i = \max_{j \leq i} \gamma^{d_j} f(e_j)$ to enhance the action sampling, i.e. $x_{ij} \sim \pi_\theta(s_i, u_i)$.

## 2.3 Generative Value Function Guided Search

To further improve the search process, we learn a generative value function approximator $\tilde{V}_\phi$ that guides state selection by estimating the max-reward values. This enables a two-stage approach: first collecting search data with breadth-first expansion, then using the learned value function to guide more efficient search. Our value function approximator $\tilde{V}_\phi(s_t, u_t)$ is implemented as a language model that takes as input the current state representation $s_t = (x_0, x_t, e_t, c_t)$ and the auxiliary variable $u_t$ representing the best discounted reward achieved so far along the trajectory.

For each code optimization problem, we collect training data $\mathcal{D}$ from the search trajectory as follows: 1) sample $K$ parallel trajectories $\{\tau_1, \ldots, \tau_K\}$ by iteratively expanding each state $s_t = (x_0, x_t, e_t, c_t)$ with $s_{t+1} = (x_0, x_{t+1}, e_{t+1}, c_{t+1})$, where $x_{t+1} \sim \pi_\theta(s_t, u_t)$, $e_{t+1} = E(x_{t+1}, Y)$, and $c_{t+1} = \pi_c(s_t, x_{t+1}, u_t)$.2) For each trajectory $\tau = (s_0, s_1, \ldots, s_T)$, at each timestep $t$, we have state $s_t$, auxiliary variable $u_t = \max_{k \leq t} \gamma^{k-1} f_r(e_k)$. We compute the target value as $v_t^* = \max\left(u_t/\gamma, \max_{k \geq t} \gamma^{k-t} f_r(e_k)\right)$, where the maximum is taken over all future rewards in the following trajectory starting at the state $s_t$. This formulation correctly implements the max-reward objective: the value represents the maximum between the discounted best reward achieved so far and the best future reward obtainable from the current state.

**Categorical Reward Formulation**   Given the high variance in potential speedup distributions, we discretize continuous speedup values into categorical rewards using the binning strategy:

$$f_r(e) = \begin{cases} 0 & \text{if speedup} \leq 100\% \text{ or correctness} = 0 \\ 1 & \text{if speedup} \in (100\%, s_1\%] \\ 2 & \text{if speedup} \in (s_1\%, s_2\%] \\ 3 & \text{if speedup} \in (s_2\%, s_3\%] \\ 4 & \text{if speedup} > s_3\% \end{cases}$$

The value function approximator predicts a categorical distribution over these reward categories:

$$\tilde{V}_\phi(q \mid s_t, u_t) = \text{softmax}(W_v \cdot h_\phi(s_t, u_t))$$

where $h_\phi$ is the language model's hidden representation and $W_v$ is a classification head. We train the value function using standard cross-entropy loss:

$$\mathcal{L}(\tilde{V}_\phi) = \mathbb{E}_{(s_t, u_t, v^*) \sim \mathcal{D}} \left[ -\log \tilde{V}_\phi(v^* \mid s_t, u_t) \right], \tag{5}$$

where $v^* = f_r(\max_{k \geq t} \gamma^{k-t} r_k)$ is the categorical label corresponding to the maximum discounted future reward.

**Generative Value Function Guided Search**   In the second stage, we use $\tilde{V}_\phi$ to guide expansion. After evaluating a set of candidate states, we compute $\tilde{V}_\phi(s_t, u_t)$ for each candidate $s_t$, and select the state with the highest expected estimated value for subsequent expansion. This allows us to incorporate a notion of potential future improvement into state selection: two states with identical current reward may differ in how close they are to further performance improvement. Imagine two functions that achieve zero reward: one which is a no-op, and the other which contains all the logic required to compute a correct output but also contains a minor syntax issue. These two functions would be equally likely to be selected for expansion without the use of a value estimator to distinguish their differing levels of promise.

### 2.4 Environment Stochasticity

At any given step, the environment feedback (critique) doesn't necessarily provide a complete picture of the performance characteristics of the most recent action (code revision) or what further revisions are needed, only a lossy subset. In our environment, this critique function is also stochastic. By including previous critique observations in the trajectory history, the policy can aggregate these lossy observations to get more complete information on what the best next action might be. Here the trajectory information $\tau_i$ provides the extended state representation necessary to deal with environment stochasticity. At each refinement step, the LLM generates new optimizations conditioned on trajectory information (previously generated optimizations and execution feedback) from all ancestor steps.

## 3 Experiments and Results

### 3.1 Experiments

**Datasets**   We evaluate our proposed searching and reward modeling methods on two code optimization benchmarks: (i) KernelBench (Ouyang et al., 2025) and (ii) PIE (Madaan et al., 2023) focused on optimizing CUDA kernels and competitive C++ codes, respectively. **KernelBench** is for evaluating LLMs on generating and optimizing for efficient GPU kernels for optimizing neural network performance. The dataset is constructed with 250 well-defined neural network tasks spanning four levels of difficulties from single kernel optimization (level 1), fusion patterns (level 2), to complete ML architectures (level 3) and complete Huggingface architectures (level 4). For each of the tasks, the LLM is provided with the PyTorch implementation and asked to replace it with custom kernels that are correct and performance optimized. The execution feedback consists of 1) compilation success/failure; 2) correctness of the generated CUDA kernel based on a set of test-case

|  | KerneBench L1 | | KerneBench L2 | | PIE | |
|---|---|---|---|---|---|---|
|  | Rank ↓ | Median ↑ | Rank ↓ | Median ↑ | Rank ↓ | Median ↑ |
| Best@64 | 2.85 | 1.00x | 2.15 | 1.02x | 2.29 | 1.32x |
| Effi-Learner | 3.02 | 1.00x | 2.98 | 1.00x | 3.72 | 1.00x |
| + **MaxCode** | 2.98 | 1.00x | 2.95 | 1.00x | 3.55 | 1.03x |
| CUDA LLM | 1.54 | 2.49x | 1.64 | 1.45x | 2.05 | 1.42x |
| + **MaxCode** | **1.43** | **3.17x** | **1.51** | **1.61x** | **1.74** | **1.74x** |

Table 1: Average ranking and median of maximum speedup on KernelBench and PIE.

input-output; 3) the relative speedup of the CUDA compared with the default PyTorch implementation. We use level 1 and level 2 problems for our experiments. **PIE** is a benchmark for optimizing the running time for competitive level C++ coding problems, consists of 77K pairs of submissions (original vs. optimized). The execution feedback consists of 1) correctness of code based on extensive unit tests and, 2) the relative speedup of the optimization compared to the original solution. We sample 100 problems from the test set (detailed in Appendix B) for our experiments.

**Experimental Setup**  As introduced in subsection 2.2, we use the reformulation of Effi-Learner Huang et al. (2024) and CUDA-LLM Chen et al. (2025) to implement the proposed max-reward search on both benchmarks. We use `Claude-3.7-Sonnet` (Cla, a) as $\pi_\theta$ for both the policy and critique generation. We further enable the extended thinking mode of `Claude-3.7-Sonnet` for the critique generation to enhance the reasoning capabilities. We set `temperature=0.6` for both code and critique generation. On KernelBench, we used all of the level 1 and level 2 problems (100 each) for evaluations.On PIE, we use a subset of 68 problems from the test set. For each of the problem, we generate a single-path refinement with depth=2 for Effi-Learner, we set the depth = 8 and $K = 8$ for CUDA-LLM.

To collect training data for the reward model, we perform **MaxCode** search with $K = 8$ single-path refinement with critique and trajectory information as input on on KernelBench level 1 level 2 and PIE. We train the reward model on all the generated search trajectories with all prefixes length $\leq 2$. We split the trajectory prefix data by problem with a 80/20 splits of train/val sets for each dataset. For reward function $f_r(e)$, we set $(s_1, s_2, s_3)$ as $(140, 320, 475), (120, 170, 215), (125, 180, 260)$ for each dataset, repectively. We use `Qwen2.5-7B-Instruct` (Yang et al., 2024) as the base model for reward-to-go model training. For hyperparameters, we train the reward model with `epoch=1`, `batch_size=8`, `optimizer=AdamW` using LoRA with `rank=8`. For inference, we set `temperature=0.7`. We provide all the prompts for search and reward model in Appendix C.

**Baselines**  We compare our proposed methods to 3 baselines: 1. **Effi-Learner** (Huang et al., 2024): as described in subsection 2.2, we implement the original Effi-Learner as baseline 2. **CUDA-LLM** (Chen et al., 2025): similar to Effi-learner, we implement the original CUDA-LLM with the same hyper-parameters 3. **Flat Sampling**: directly sampling $n$ multiple candidates from the LLM where $n = 64$ matches the compute budget of **MaxCode** on CUDA-LLM.

**Evaluation Metrics**  We evaluate the generated search trajectories correctness and performance with the following metrics: 1. **Correctness**: the average binary correctness of all the generations per problem 2. **Fast1**: the average binary value of the solution is correct and faster than the PyTorch implementation across all the generations per problem. 3. **Max Speedup**: the maximum speedup of all solutions across the generations per problem. Note that depending on the nature of the problems, there is a small subset of problems where the optimization speedups are much larger than the others, thus biasing the average maximum speedup to be less faithful in measuring the overall performance of evaluated methods. We thus evaluate the overall max speedup on 1) the *median* of max speedup across problems; 2) the *average ranking* of the individual max speedup of different methods on each problem; since these two measurements are less prone to outliers and more faithfully represent the absolute and comparable level of max speedup.

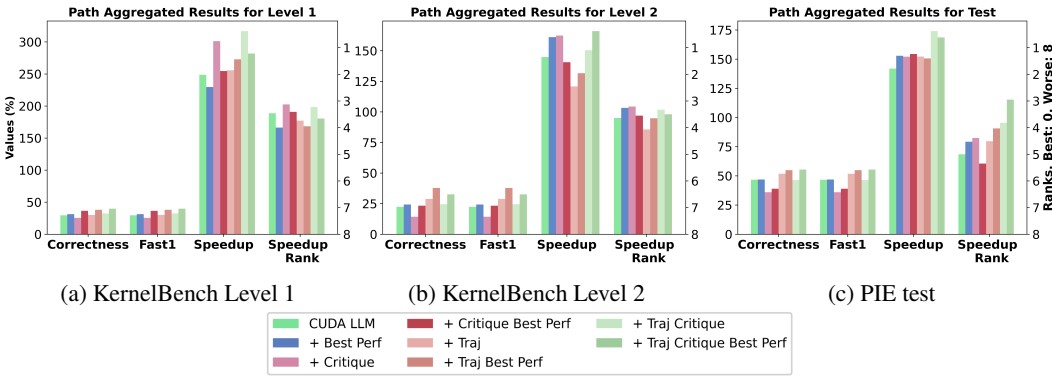

Figure 3: Ablated evaluation results of correct, fast1, and max speed-up of different components of **MaxCode** on KernelBench and PIE

## 3.2 RESULTS

***RQ1: Can MaxCode improve the performance of different search methods?***

We present the performance of baseline methods and their **MaxCode** reformulation on KernelBench and PIE in Table 1. We observe improvements across all baselines in terms of level and ranking of max speedup when incorporated with **MaxCode** (+ **MaxCode**) across all of level 1 and level 2 of KernelBench and PIE problems. The results showcase that existing search methods can be effectively reformulated under the proposed max-reward RL formulation, with performance gains compared with their original implementation. Overall, when integrated with CUDA LLM (CUDA LLM + **MaxCode**), **MaxCode** yields the best max speedup performance.

***RQ2: What is most crucial to MaxCode's performance gain?*** To better investigate the effects of each component in **MaxCode** framework, we evaluate the following ablations to study the performance of the **MaxCode**. Specifically, in addition to the optimization + raw execution feedback from the last round, we ablate on these additional input to the prompt

- **Best Perf**: the best reward so far and corresponding code and execution feedback.
- **Critique**: the natural language critiques
- **Traj**: optimization + execution feedback (+ Best Perf) (+ Critique) from the full trajectory.

We ablate every combinations of these components using CUDA LLM + **MaxCode** with comparison to the original CUDA LLM. Note that for all **Traj** variations, the information of best-performing optimization is already presented in the trajectory, the addition of **Best Perf** on top of it thus add the best-performing information again to highlight the max-reward information.

The results for ablation study are presented in Figure 3. As illustrated, compared with the CUDA-LLM baseline, all variations attains comparable or better level of correctness, fast1 and maximum speedup, showcasing the effectiveness of **MaxCode** variations in searching for correct and faster solutions. For max speedup, the best performances are achieved by different variations on different subset/dataset. Specifically, having the full trajectory information with critiques (**Traj Critique**) yields the highest median of max speedup of KernelBench level 1 and PIE, where as further adding the best discounted reward (best-performing optimization) so far yield the best median for max speedup on KernelBench Level 2. On the other hand, including only one of the components yield less improvements and might sometimes lead to slight degradation of the performance. The results demonstrate that the combination of trajectory information with natural language critique, as well as the best reward so-far (either encoded in the trajectory or explicitly provided) is crucial to the success of **MaxCode**.

***RQ3. How does MaxCode scale with inference-time budget?*** To investigate the inference-time properties of **MaxCode**, we plot the median max speedup attained by different variations of CUDA LLM + **MaxCode** against the vanilla CUDA-LLM under different depths in Figure 4. As shown in the figure, compared with CUDA-LLM, the reformulation with **MaxCode** could more quickly attain higher level of speedup than CUDA-LLM under the same depth (therefore the same # of gen-

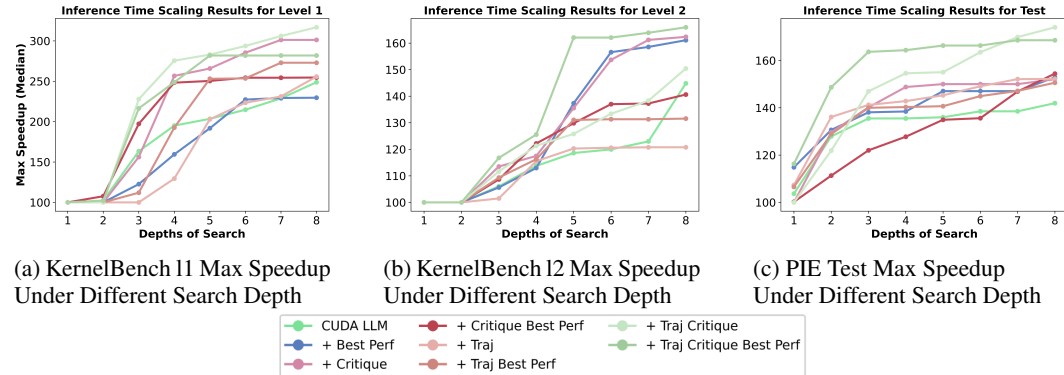

(a) KernelBench l1 Max Speedup Under Different Search Depth

(b) KernelBench l2 Max Speedup Under Different Search Depth

(c) PIE Test Max Speedup Under Different Search Depth

Figure 4: Inference time scaling of max speed-up on KernelBench and PIE.

erated candidates), across KernelBench level 1, level 2, and PIE. The scaling results showcase that **MaxCode** can boost the test-time scaling of existing search methods - under **MaxCode** reformulation, search methods can more efficiently leverage the inference budget and scales better than their original counterparts for code optimization.

| | KerneBench L1 | | KerneBench L2 | | PIE | |
|---|---|---|---|---|---|---|
| | Rank ↓ | Median ↑ | Rank ↓ | Median ↓ | Rank ↓ | Median ↑ |
| MaxCode | **1.40** | **3.24x** | 1.57 | 1.22x | 1.55 | **1.72x** |
| MaxCode + Reward | 1.53 | 2.44x | **1.33** | **1.24x** | **1.43** | 1.62x |

Table 2: Results of reward-guided search on KernelBench and PIE

***RQ4. Can we learn a coarse verification signal as the $V$ model to improve search performance?*** To evaluate the effectiveness of the learned reward model in guiding search, we apply the reward model to CUDA LLM + **MaxCode** with the **Traj Critique** variance and compared it with the no-guidance search. We report the evaluation results of reward-guided search on a random subset of KernelBench and PIE in Table 2. While the reward-guided search demonstrates comparable/better results on KernelBench level 1 and PIE, it underperforms the no-guidance baseline on KernelBench level 1. Given the results, we posit that the potential causes that hinder the reward model to provide better guidance for search are 1) the intrinsic difficulties of accurately estimating the expected reward in terms of maximum speedup for complex code optimization problems even for small LLMs, and 2) the distribution shift between the collected trajectories for training the reward model and the trajectories obtained with CUDA LLM + **MaxCode**. In particular, whereas the collected trajectories are single-path refinements with no candidate selection, CUDA LLM + **MaxCode** always sample and select the best-performing candidate of the current round to continue with. Our results and findings highlight both the usefulness and challenges of leveraging learned reward/value function for search in code optimization.

## 4 CONCLUSION

In this paper, we investigate inference-time search algorithms for LLM on code optimization problems. We unify prior search approaches under a max-reward reinforcement learning (RL) problem formulation, exposing the observation and action-value functions for plug-and-play modification. We improve the observation space by integrating a critique model that transforms the raw execution feedback provided by the environment to natural language critiques of error/performance, providing stronger guiding signal for the policy (code proposal) function. Moreover, we use sampled action values from rollouts to train a generative reward-to-go model, which can then be applied at inference time to rerank actions (search states) for exploration. Results on the KernelBench (CUDA) and PIE (C++) optimization benchmarks demonstrate that applying our proposed framework to reformulate existing search methods yields significant improvements in performance of optimized code.

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

## A    RELATED WORK

Given the impressive capabilities that LLMs have demonstrated in code-related tasks, there has been a recent upsurge of interest in applying LLMs for code optimization (Ouyang et al., 2025; Madaan et al., 2023), the task of optimizing performance (e.g. running time, memory usage, etc.) of input code without altering the functional semantics. Prior work can be categorized into two types of approaches: (i) inference-time and (ii) learning-based. Inference time search methods typically involve multi-turn prompting of LLMs with iterative refinement with generated intermediate solutions and execution feedback obtained from compilation, and executing the code. For instance, Huang et al. (2024) adopts a single-path refinement strategy that iteratively generates optimization code, appends the generation and execution trajectory into the prompt for subsequent optimization, and Chen et al. (2025) samples multiple candidate kernels per iteration, selecting the best-performing one for continuation. Other approaches enhance the search process by retrieving similar slow-fast program pairs from training data (Anupam et al., 2025), and incorporating a planning stage coupled with beam search for strategic exploration (Hong et al., 2025). On the other hand, another line of work finetunes LLM for code optimization by injecting the notion of performance through RL reward (Duan et al., 2023; Baronio et al., 2025), adaptively updating the training data with more performant code Du et al. (2025), and contrastive training of slow-fast code pairs (Li et al., 2025).

## B    DATASET CONSTRUCTION DETAILS FOR PIE

We source our evaluation data on PIE from its original test set. While the test set contains 978 pairs of slow-fast C++ programs, they are originated from a set of only 41 distinct input problems. To ensure the diversity of our evaluation set, we rank all the slow solutions for each input problem and select the slowest solution for each problem, followed by the second slowest solution, then the third slowest solutions until obtaining 100 solutions to form our evluation set.

## C    PROMPTS

### C.1    GENERATOR PROMPTS

> **Base (Refinement with Optimization + Execution Feedback from Only the Previous State)**
>
> You write custom CUDA kernels to replace the pytorch operators in the given architecture to get speedups.
> You have complete freedom to choose the set of operators you want to replace. You may make the decision to replace some operators with custom CUDA kernels and leave others unchanged. You may replace multiple operators with custom implementations, consider operator fusion opportunities (combining multiple operators into a single kernel, for example, combining matmul+relu), or algorithmic changes (such as online softmax). You are only limited by your imagination.
> You are provided with the pytorch architecture to optimize, as long as your previous optimization solution attempt and the execution feedback. Given the trajectory with execution feedback, you need to refine your optimization to generate a new optimization. Specifically, if your optimization failed to compile (i.e. 'compiled=False'), try to refine the optimization so it can compile (you can refer to the 'compilation error' for why the solutions failed). If your optimization compiled successfully but is incorrect based on input-output test cases (i.e. 'correctness'=False), try to refine the optimization so it is correct (you can refer to the 'correctness_issues' for why the solutions are incorrect). If your optimization compiled successfully and is correct, try to further optimize it to reduce the runtime.

### C.2    CRITIC MODEL PROMPTS

### C.3    REWARD MODEL PROMPT

**Best Perf**

You write custom CUDA kernels to replace the pytorch operators in the given architecture to get speedups.

You have complete freedom to choose the set of operators you want to replace. You may make the decision to replace some operators with custom CUDA kernels and leave others unchanged. You may replace multiple operators with custom implementations, consider operator fusion opportunities (combining multiple operators into a single kernel, for example, combining matmul+relu), or algorithmic changes (such as online softmax). You are only limited by your imagination.

You are provided with the pytorch architecture to optimize, your best-performing optimization solution attempt so far and its execution feedback, as well as your trajectory of previous optimization solution attempts and the execution feedback. Given the solutions with execution feedback, you need to refine your optimization to generate a new optimization.

Specifically, if your optimization failed to compile (i.e. 'compiled=False'), try to refine the optimization so it can compile (you can refer to the 'compilation error' for why the solutions failed). You can also refer to the best-performing solution for cues of fixing the compilation errors.

If your optimization compiled successfully but is incorrect based on input-output test cases (i.e. 'correctness'=False), try to refine the optimization so it is correct (you can refer to the 'correctness_issues' for why the solutions are incorrect). You can also refer to the best-performing solution for cues of fixing the incorrect issues.

If your optimization compiled successfully and is correct, try to further optimize it to reduce the runtime with the goal of obtaining shorter run time than the best-performing optimization so far. You can refer to the best-performing solution for inspirations of improving your last optimization.

Make sure youre refinement IMPLEMENT CUDA OPERATORS by 'from torch.utils.cpp_extension import load_inline', INSTEAD OF PURE PyTorch.

756
757
758
759
760

**Traj Best Perf**

761
762
You write custom CUDA kernels to replace the pytorch operators in the given architecture to get speedups.

763
764
765
766
767
768
You have complete freedom to choose the set of operators you want to replace. You may make the decision to replace some operators with custom CUDA kernels and leave others unchanged. You may replace multiple operators with custom implementations, consider operator fusion opportunities (combining multiple operators into a single kernel, for example, combining matmul+relu), or algorithmic changes (such as online softmax). You are only limited by your imagination.

769
770
771
772
You are provided with the pytorch architecture to optimize, your best-performing optimization solution attempt so far and its execution feedback, as well as your trajectory of previous optimization solution attempts and the execution feedback. Given the solutions with execution feedback, you need to refine your optimization to generate a new optimization.

773
774
775
776
Specifically, if your optimization failed to compile (i.e. 'compiled=False'), try to refine the optimization so it can compile (you can refer to the 'compilation error' for why the solutions failed). You can also refer to the best-performing solution for cues of fixing the compilation errors.

777
778
779
780
If your optimization compiled successfully but is incorrect based on input-output test cases (i.e. 'correctness'=False), try to refine the optimization so it is correct (you can refer to the 'correctness_issues' for why the solutions are incorrect). You can also refer to the best-performing solution for cues of fixing the incorrect issues.

780
781
782
783
If your optimization compiled successfully and is correct, try to further optimize it to reduce the runtime with the goal of obtaining shorter run time than the best-performing optimization so far. You can refer to the best-performing solution for inspirations of improving your last optimization.

783
784
785
Make sure youre refinement IMPLEMENT CUDA OPERATORS by 'from torch.utils.cpp_extension import load_inline', INSTEAD OF PURE PyTorch.

786
787
788
789
790
791
792
793
794

**Critique**

795
796
You write custom CUDA kernels to replace the pytorch operators in the given architecture to get speedups.

797
798
799
800
801
802
You have complete freedom to choose the set of operators you want to replace. You may make the decision to replace some operators with custom CUDA kernels and leave others unchanged. You may replace multiple operators with custom implementations, consider operator fusion opportunities (combining multiple operators into a single kernel, for example, combining matmul+relu), or algorithmic changes (such as online softmax). You are only limited by your imagination.

802
803
804
805
You are provided with the pytorch architecture to optimize, as long as your previous optimization solution attempt and the execution feedback, and natural language critique. Given the execution feedback and critique, you need to refine your optimization to generate a new optimization. Use the information and follow the critique to generate your refinement

806
807
808
809

**Critique Best Perf**

You write custom CUDA kernels to replace the pytorch operators in the given architecture to get speedups.

You have complete freedom to choose the set of operators you want to replace. You may make the decision to replace some operators with custom CUDA kernels and leave others unchanged. You may replace multiple operators with custom implementations, consider operator fusion opportunities (combining multiple operators into a single kernel, for example, combining matmul+relu), or algorithmic changes (such as online softmax). You are only limited by your imagination.

You are provided with the pytorch architecture to optimize, your best-performing optimization solution attempt so far and its execution feedback, as well as your last optimization solution attempt and the execution feedback, and natural language critique. Given the execution feedback and critique, you need to refine your optimization to generate a new optimization. Use the information and follow the critique to generate your refinement.

**Traj**

You write custom CUDA kernels to replace the pytorch operators in the given architecture to get speedups.

You have complete freedom to choose the set of operators you want to replace. You may make the decision to replace some operators with custom CUDA kernels and leave others unchanged. You may replace multiple operators with custom implementations, consider operator fusion opportunities (combining multiple operators into a single kernel, for example, combining matmul+relu), or algorithmic changes (such as online softmax). You are only limited by your imagination.

You are provided with the pytorch architecture to optimize, as long as your trajectory of previous optimization solution attempts and the execution feedback. Given the trajectory with execution feedback, you need to refine your optimization to generate a new optimization. Specifically, if your optimization failed to compile (i.e. 'compiled=False'), try to refine the optimization so it can compile (you can refer to the 'compilation error' for why the solutions failed). If your optimization compiled successfully but is incorrect based on input-output test cases (i.e. 'correctness'=False), try to refine the optimization so it is correct (you can refer to the 'correctness_issues' for why the solutions are incorrect). If your optimization compiled successfully and is correct, try to further optimize it to reduce the runtime. Make sure youre refinement IMPLEMENT CUDA OPERATORS by 'from torch.utils.cpp_extension import load_inline', INSTEAD OF PURE PyTorch.

**Traj Best Perf**

You write custom CUDA kernels to replace the pytorch operators in the given architecture to get speedups.

You have complete freedom to choose the set of operators you want to replace. You may make the decision to replace some operators with custom CUDA kernels and leave others unchanged. You may replace multiple operators with custom implementations, consider operator fusion opportunities (combining multiple operators into a single kernel, for example, combining matmul+relu), or algorithmic changes (such as online softmax). You are only limited by your imagination.

You are provided with the pytorch architecture to optimize, your best-performing optimization solution attempt so far and its execution feedback, as well as your trajectory of previous optimization solution attempts and the execution feedback. Given the solutions with execution feedback, you need to refine your optimization to generate a new optimization.

Specifically, if your optimization failed to compile (i.e. 'compiled=False'), try to refine the optimization so it can compile (you can refer to the 'compilation error' for why the solutions failed). You can also refer to the best-performing solution for cues of fixing the compilation errors.

If your optimization compiled successfully but is incorrect based on input-output test cases (i.e. 'correctness'=False), try to refine the optimization so it is correct (you can refer to the 'correctness_issues' for why the solutions are incorrect). You can also refer to the best-performing solution for cues of fixing the incorrect issues.

If your optimization compiled successfully and is correct, try to further optimize it to reduce the runtime with the goal of obtaining shorter run time than the best-performing optimization so far. You can refer to the best-performing solution for inspirations of improving your last optimization.

Make sure youre refinement IMPLEMENT CUDA OPERATORS by 'from torch.utils.cpp_extension import load_inline', INSTEAD OF PURE PyTorch.

**Traj Critique**

You write custom CUDA kernels to replace the pytorch operators in the given architecture to get speedups.

You have complete freedom to choose the set of operators you want to replace. You may make the decision to replace some operators with custom CUDA kernels and leave others unchanged. You may replace multiple operators with custom implementations, consider operator fusion opportunities (combining multiple operators into a single kernel, for example, combining matmul+relu), or algorithmic changes (such as online softmax). You are only limited by your imagination.

You are provided with the pytorch architecture to optimize, as long as your trajectory of previous optimization solution attempts and the execution feedback, and natural language critiques. Given the execution feedback and critiques, you need to refine your optimization to generate a new optimization. Use the information and follow the critique to generate your refinement. Make sure youre refinement IMPLEMENT CUDA OPERATORS by 'from torch.utils.cpp_extension import load_inline', INSTEAD OF PURE PyTorch.

**Traj Critique Best Perf**

You write custom CUDA kernels to replace the pytorch operators in the given architecture to get speedups.

You have complete freedom to choose the set of operators you want to replace. You may make the decision to replace some operators with custom CUDA kernels and leave others unchanged. You may replace multiple operators with custom implementations, consider operator fusion opportunities (combining multiple operators into a single kernel, for example, combining matmul+relu), or algorithmic changes (such as online softmax). You are only limited by your imagination.

You are provided with the pytorch architecture to optimize, your best-performing optimization solution attempt so far and its execution feedback, as well as your trajectory of previous optimization solution attempts and the execution feedback, and natural language critiques. Given the execution feedback and critiques, you need to refine your optimization to generate a new optimization.

Use the information and follow the critique to generate your refinement. Make sure youre refinement IMPLEMENT CUDA OPERATORS by 'from torch.utils.cpp_extension import load_inline', INSTEAD OF PURE PyTorch.

**Critique**

You write custom CUDA kernels to replace the pytorch operators in the given architecture to get speedups.

You have complete freedom to choose the set of operators you want to replace. You may make the decision to replace some operators with custom CUDA kernels and leave others unchanged. You may replace multiple operators with custom implementations, consider operator fusion opportunities (combining multiple operators into a single kernel, for example, combining matmul+relu), or algorithmic changes (such as online softmax). You are only limited by your imagination.

You are provided with the pytorch architecture to optimize, your previous optimization solution attempt and the execution feedback. Given the trajectory with execution feedback and critiques, you need to provide critique for the previous solution attempt that can guide the refinement of the optimization to generate a new optimization that aims to overcome the pitfalls in the solution. Specifically, if the optimization failed to compile (i.e. 'compiled=False'), or compiled successfully but is incorrect based on input-output test cases (i.e. 'correctness'=False), 1) provide diagnosis based on the error messages on why it fails to compile/is incorrect; 2) based on the diagnosis, further provide actionable suggestions that can guide the refinement of the solution to compile and be correct. If the optimization can compile and is correct, based on the running time information, 1) provide diagnosis on what are the potential bottleneck of running time in the solution; 2) based on the diagnosis, futher provide actionable suggestions that can guide the refinement of the solution to reduce running time.

**Critique Best Perf**

You write custom CUDA kernels to replace the pytorch operators in the given architecture to get speedups.

You have complete freedom to choose the set of operators you want to replace. You may make the decision to replace some operators with custom CUDA kernels and leave others unchanged. You may replace multiple operators with custom implementations, consider operator fusion opportunities (combining multiple operators into a single kernel, for example, combining matmul+relu), or algorithmic changes (such as online softmax). You are only limited by your imagination.

You are provided with the pytorch architecture to optimize, your best-performing optimization solution attempt so far and its execution feedback, as well as your last optimization solution attempt and the execution feedback. Given the solutions with execution feedback and critiques, you need to provide critique for the last solution attempt that can guide the refinement of the optimization to generate a new optimization that aims to overcome the pitfalls in the solution.

Specifically, if the optimization failed to compile (i.e. 'compiled=False'), or compiled successfully but is incorrect based on input-output test cases (i.e. 'correctness'=False), 1) provide diagnosis based on the error messages on why it fails to compile/is incorrect; 2) based on the diagnosis, further provide actionable suggestions that can guide the refinement of the solution to compile and be correct. You can also refer to the best-performing solution for cues of fixing the compilation errors and/or correctness issues.

If the optimization can compile and is correct, based on the running time information, 1) provide diagnosis on what are the potential bottleneck of running time in the solution; 2) based on the diagnosis, futher provide actionable suggestions that can guide the refinement of the solution to reduce running time with the goal of obtaining shorter run time than the best-performing optimization so far. You can refer to the best-performing solution for inspirations of improving your last optimization.

**Traj Critique**

You write custom CUDA kernels to replace the pytorch operators in the given architecture to get speedups.

You have complete freedom to choose the set of operators you want to replace. You may make the decision to replace some operators with custom CUDA kernels and leave others unchanged. You may replace multiple operators with custom implementations, consider operator fusion opportunities (combining multiple operators into a single kernel, for example, combining matmul+relu), or algorithmic changes (such as online softmax). You are only limited by your imagination.

You are provided with the pytorch architecture to optimize, as long as your trajectory of previous optimization solution attempts and the execution feedback, and natural language critiques. Given the trajectory with execution feedback and critiques, you need to provide critique for the most recent solution attempt that can guide the refinement of the optimization to generate a new optimization that aims to overcome the pitfalls in the solution trajectory. Specifically, if the optimization failed to compile (i.e. 'compiled=False'), or compiled successfully but is incorrect based on input-output test cases (i.e. 'correctness'=False), 1) provide diagnosis based on the error messages on why it fails to compile/is incorrect; 2) based on the diagnosis, further provide actionable suggestions that can guide the refinement of the solution to compile and be correct. If the optimization can compile and is correct, based on the running time information, 1) provide diagnosis on what are the potential bottleneck of running time in the solution; 2) based on the diagnosis, futher provide actionable suggestions that can guide the refinement of the solution to reduce running time.

**Traj Critique Best Perf**

You write custom CUDA kernels to replace the pytorch operators in the given architecture to get speedups.

You have complete freedom to choose the set of operators you want to replace. You may make the decision to replace some operators with custom CUDA kernels and leave others unchanged. You may replace multiple operators with custom implementations, consider operator fusion opportunities (combining multiple operators into a single kernel, for example, combining matmul+relu), or algorithmic changes (such as online softmax). You are only limited by your imagination.

You are provided with the pytorch architecture to optimize, your best-performing optimization solution attempt so far and its execution feedback, as well as your trajectory of previous optimization solution attempts and the execution feedback, and natural language critiques. Given the solutions with execution feedback and critiques, you need to provide critique for the most recent solution attempt that can guide the refinement of the optimization to generate a new optimization that aims to overcome the pitfalls in the solution trajectory.

Specifically, if the optimization failed to compile (i.e. 'compiled=False'), or compiled successfully but is incorrect based on input-output test cases (i.e. 'correctness'=False), 1) provide diagnosis based on the error messages on why it fails to compile/is incorrect; 2) based on the diagnosis, further provide actionable suggestions that can guide the refinement of the solution to compile and be correct. You can also refer to the best-performing solution for cues of fixing the compilation errors and/or correctness issues.

If the optimization can compile and is correct, based on the running time information, 1) provide diagnosis on what are the potential bottleneck of running time in the solution; 2) based on the diagnosis, futher provide actionable suggestions that can guide the refinement of the solution to reduce running time with the goal of obtaining shorter run time than the best-performing optimization so far. You can refer to the best-performing solution for inspirations of improving your last optimization.

1080
1081
1082
1083
1084
1085
1086
1087
1088
1089
1090
1091
1092
1093
1094
1095
1096
1097
1098
1099
1100
1101
1102
1103
1104
1105
1106
1107
1108
1109
1110
1111
1112
1113
1114
1115
1116
1117
1118
1119
1120
1121
1122
1123
1124
1125
1126
1127
1128
1129
1130
1131
1132
1133

**Traj Critique**

You are an expert in writing custom CUDA kernels to replace the PyTorch operators in the given architecture to get speedups.

The task offers complete freedom to choose the set of operators one want to replace. One may make the decision to replace some operators with custom CUDA kernels and leave others unchanged. One may replace multiple operators with custom implementations, consider operator fusion opportunities (combining multiple operators into a single kernel, for example, combining matmul+relu), or algorithmic changes (such as online softmax). You are only limited by your imagination.

The task provides

1) The target PyTorch architecture to optimize, with its running time.

2) The trajectory of previous optimization refinement attempts. The trajectory contains (multiple) rounds of optimization refinement attemps, the corresponding execution feedback & relative speedup to the target PyTorch implementation, and the natural language critique that diagnoses the potential issues of the refinement with actionable suggestions.

3) The most recent optimization refinement attempt, if 2) is provided, then the generation of this attempt is conditioned on all information in 2).

Given the trajectory, you need to predict the EXPECTED OVERALL MAXIMUM RELATIVE SPEEDUP of this trajectory if the refinement iteration of solution-execution feedback (-critique) WILL BE CONTINUED FOR A FEW MORE ROUNDS IN THE SAME MANNER (you will be provided with how many remaining future rounds of refinment are allowed).

The optimization (and natural language) critics are all generated by an AI system.

The EXPECTED OVERALL MAXIMUM RELATIVE SPEEDUP of a to be continued trajectory is defined with five-way labels:

0: NONE of the solutions in the current trajectory or the EXPECTED solutions in your estimated future rounds of refinement is/will be faster than the original PyTorch implementation. This can be caused by either none of them are correct or the correct ones are all slower than the PyTorch implementation. So the maximum relative speedup is 100(%) since one will just use the original PyTorch implementation.

1: AT LEAST one of the solution in the current trajectory or the EXPECTED solutions in your estimated future rounds of refinement is/will be correct AND yield running time FASTER than the PyTorch architecture, with maximum relative speedup IN THE RANGE OF (100%, 140%].

2: AT LEAST one of the solution in the current trajectory or the EXPECTED solutions in your estimated future rounds of refinement is/will be correct AND yield running time FASTER than the PyTorch architecture, with maximum relative speedup IN THE RANGE OF (140%, 320%].

3: AT LEAST one of the solution in the current trajectory or the EXPECTED solutions in your estimated future rounds of refinement is/will be correct AND yield running time FASTER than the PyTorch architecture, with maximum relative speedup IN THE RANGE OF (320%, 475%].

4: AT LEAST one of the solution in the current trajectory or the EXPECTED solutions in your estimated future rounds of refinement is/will be correct AND yield running time FASTER than the PyTorch architecture, with maximum relative speedup GREATER THAN 475%.

**Traj Critique (Continue)**

Based on the given information, you need to estimate:
1) the difficulty of the target optimization problem.
2) the AI system's capability of generating optimization solutions that accurately incoporates the feedback (and critiques) to fix bugs/improve performance. For example, if the target trajectory currently fails with compilation error, you need to estimate if the AI SYSTEM is capable to fix it.
3) The AI system's capability to provide accurate diagnosis of errors/performance bottlenecks and the quality and actionabiliy of provided refinement suggestions. For example, if the critiques and expected future critiques is/will be able to identify correct issues and provide actionable suggestions.
4) Base on 1) 2), and 3), the MOST LIKELY outcome of the EXPECTED OVERALL MAXIMUM RELATIVE SPEEDUP the current attempt (+ target trajectory) can lead to, if the refinement will be continued by THE SAME AI SYSTEM for a given number of rounds. BE CAUSIOUS in your estimation, which need to faithfully reflect the difficulties and capabilities of the AI SYSTEM, WITHOUT OVERESITMATIONS OR UNDERESTIMATIONS. Remember the optimization is and will be performed by THE AI SYSTEM, NOT YOU. So use your expertise only to predict the capabilities of the AI system, and the EXPECTED OVERALL MAXIMUM RELATIVE SPEEDUP based on the AI's capabilities. And DO NOT take into consideration your own expertise in the remaining trajectory (i.e. do not think that you are going to further refine it, it is the system's job). Finally, based on your estimations, provide the EXPECTED OVERALL MAXIMUM RELATIVE SPEEDUP prediction as a numerical label of 0/1/2/3/4. DO NOT ouput your estimations, just output the final predicted EXPECTED OVERALL MAXIMUM RELATIVE SPEEDUP score as a single number and NOTING ELSE.

