# OpenReview forum: "MaxCode: A Max-Reward Reinforcement Learning Framework for Automated Code Optimization"
_ICLR.cc/2026/Conference — ICLR 2026 Conference Withdrawn Submission_

### Official Review · Reviewer_qxWk · 2025-10-26

**Soundness:** 3
**Presentation:** 1
**Contribution:** 2
**Rating:** 2
**Confidence:** 3

**Summary:**

This paper proposes MaxCode, a framework that casts LLM-based code optimization as a max-reward reinforcement learning problem, focusing on finding the single best-performing solution rather than maximizing cumulative rewards. They enhanced the observation space by using an LLM-based "critique" model to translate raw execution feedback (e.g., timing, errors) into actionable, natural language insights, and collect the best-seen reward so far. The framework demonstrates significant performance improvements on the KernelBench (CUDA) and PIE (C++) optimization benchmarks.

**Strengths:**

- **Actionable Feedback Loop:** Raw execution feedback (e.g., "20% slower than the baseline") is often unhelpful. Translating this into a diagnostic natural language critique (e.g., "probable memory bandwidth bottleneck, consider fusing operations") provides a much richer and more actionable signal for the generator LLM's iterative refinement.
- **Strong Empirical Results:** The method achieves significant relative speedup improvements over strong baselines. The ablation studies validated that the critique and trajectory components are the primary drivers of this success.

**Weaknesses:**

- **Critique Model as a Black Box:** The critique model is central to the paper's positive results, but it is treated as a given. There is no analysis of the quality of the critiques, common failure modes (e.g., does it ever "hallucinate" a bottleneck?), or whether a much smaller, fine-tuned, or distilled model could serve this purpose at a fraction of the cost.
- **Unanalyzed Computational Cost:** The framework introduces significant computational overhead. At each step, it requires an inference pass from a powerful generator LLM and a separate critique LLM. This effectively at least doubles the LLM-related inference cost, in addition to the already expensive code execution step. This cost-benefit trade-off is not discussed.
- **Clarity and Presentation:** The paper's clarity could be improved, making it difficult to follow. The introduction uses several non-standard terms (e.g., "max-reward inference operator," "best-discounted reward," "categorical Value model") without immediate, clear explanations. While the meaning can be inferred, a brief definition would be benficial to the readability.
- **Minor:** The presentation of the figures needs improvement. The text in Figure 1 is too small to be legible, making its purpose unclear. Similarly, Figure 2 is titled "MaxCode search method," but the diagram is a high-level data-flow chart, and it isn't immediately obvious how it represents a search algorithm.

**Questions:**

- The critique model demonstrates a strong capability for generating high-quality, actionable suggestions. Have the authors considered using this model to directly generate code refinements, rather than only using its output as an observation for the separate policy model? This could potentially simplify the overall framework.
- Could you elaborate on the training details for the reward-to-go model? The paper states it was trained on trajectory prefixes of length less than 2. For such short sequences, the max-reward value function would seem to be heavily influenced by the immediate reward rather than longer-term potential. What was the rationale for this choice, and how are the target values computed for these short prefixes when they are sampled from longer rollouts?

---

### Official Review · Reviewer_cyNi · 2025-10-31

**Soundness:** 1
**Presentation:** 1
**Contribution:** 2
**Rating:** 2
**Confidence:** 4

**Summary:**

This paper presents MaxCode, a max-reward RL formulation for inference-time code optimization. The method models iterative refinement as an MDP whose state augments initial/current code with execution feedback and natural-language critiques. It defines a max-reward return and trains a categorical reward-to-go/value model by discretizing speedup into bins; the learned value is then used to guide candidate expansion/selection at search time. Experiments on KernelBench (L1/L2) and PIE (subset) integrate MaxCode into Effi-Learner and CUDA-LLM, reporting gains in median max-speedup and average rank, with larger margins when combined with CUDA-LLM. Overall, the paper provides a unifying inference-time optimization view with modest empirical improvements.

**Strengths:**

- Problem relevance: Inference-time optimization for LLM-generated code is practical and timely; even small speedups can be impactful in deployment.
- Conceptual unification: Recasting existing refinement methods under a max-reward RL framework offers a common lens and a value-guided expansion heuristic that can plug into multiple search variants.
- Empirical signal: On KernelBench/PIE, integrating MaxCode yields consistent but incremental improvements overall, especially with CUDA-LLM.

**Weaknesses:**

- The framework mainly adapts existing iteration-based methods and execution-feedback-based prompt strategies; the max-reward RL formulation appears conceptual rather than introducing new algorithms.
- Several core elements (e.g., critique usage, value estimation mechanism, Q-function applicability) are only loosely described, limiting reproducibility and technical insight.
- Performance gains are small and inconsistent, baselines lack diversity beyond execution-feedback approaches, and ablations do not sufficiently justify which components drive improvements.
- I find the paper difficult to follow. Important technical components are underspecified (e.g., how critique feedback is incorporated into the model state, how the Q-function is estimated in practice, and the exact architecture/training details for the reward-to-go model), making the approach challenging to reproduce or fully assess.

**Questions:**

- Could you better justify how the max-reward formulation changes the decision process compared to existing iterative refinement? Currently, the policy remains fixed, so what specific improvement does RL provide beyond a unifying view?
- Could you explicitly describe how natural-language critique influences candidate generation? Is critique concatenated in prompts only, or parsed into structured signals?
- The paper lacks architecture and training details (prompt format, token budget, sampling strategy, stopping criteria). Could you elaborate to ensure reproducibility?
- Since Effi-Learner was originally evaluated on Effi-Bench, why is Effi-Bench not included here? Would broader and more standard evaluation benchmarks affect conclusions?
- Search-based optimization is stochastic. Could you report variance/error bars and test significance to confirm improvements are reliable?
- The proposed method increases search iterations and critique calls. What is the runtime overhead relative to the speedup gains? Does the improvement justify the added compute cost?·

---

### Official Review · Reviewer_drvW · 2025-11-03

**Soundness:** 3
**Presentation:** 2
**Contribution:** 2
**Rating:** 4
**Confidence:** 3

**Summary:**

The authors propose MaxCode, a framework that formulates LLM code optimization as a max-reward RL problem. Its core idea is to augment the observation space using two key components: the best-so-far reward and a natural language critique generated by a separate LLM. This critique model translates raw execution feedback into actionable, diagnostic insights. Experiments show that the MaxCode framework significantly improves the optimization performance of existing search methods on the KernelBench and PIE benchmarks.

**Strengths:**

-	The paper tackles optimizing code efficiency via LLMs problem by reframing inference-time search under a unified max-reward RL perspective.

-	The formulation is modular and general, making it easy to plug into existing search pipelines such as CUDA-LLM or Effi-Learner, with consistent empirical gains across both CUDA and C++ domains.

-	The critique-augmented observation design is intuitive yet effective, improving exploration quality without modifying base model weights.

-	The experimental section covers multiple realistic benchmarks (KernelBench, PIE) and provides reasonably detailed ablations that support the main claims.

**Weaknesses:**

-	The proposed max-reward RL formulation mainly reinterprets existing search heuristics under a unified lens. While clean and modular, it does not introduce a fundamentally new learning algorithm or search operator beyond the combination of best-so-far reward and critique-based observation.

-	The section describing the generative value/reward-to-go model contradicts its reported results: the text claims underperformance on KernelBench-L1, yet Table 2 shows a clear gain. This discrepancy weakens confidence in the analysis and leaves the effectiveness of the reward model unclear.

-	The “distribution-shift” explanation for the reward model’s instability is plausible but untested. No evidence (e.g., off-policy evaluation or correlation between predicted and actual rewards) is provided to substantiate this claim.

-	Figures are visually cluttered, lack clear labels or legends,  and do not convey the intended insights.

**Questions:**

-	Can you clarify the contradiction between the textual description of RQ4 and Table 2? Was this a labeling or analysis error?

-	Does the critique model remain effective if used with open-source policies (e.g., CodeLlama, Qwen2-Coder), or is it tightly coupled to Claude-Sonnet?


-	Have you quantified how well the reward-to-go predictions correlate with actual speedup or final reward?

---

### Official Review · Reviewer_2hbN · 2025-11-04

**Soundness:** 2
**Presentation:** 3
**Contribution:** 2
**Rating:** 2
**Confidence:** 3

**Summary:**

The authors point out two challenging when optimizing code, including the complexity of code writing and the interpretation of performance metrics. To solve them, they explore inference-time search algorithms and propose the MaxCode framework, which unifies existing search methods under a max-reward reinforcement learning. The experiments on CUDA and C++ optimization benchmarks demonstrate the efficiency of the proposed method.

**Strengths:**

1. The topic and direction are practical and promising. Code optimization is important for CUDA and C++.
2. The paper is well-written, and the motivation is clear.

**Weaknesses:**

1. The presentation needs to be improved, especially Figure 1. The code in the figure is not clear.
2. Commas and periods are missing in the formulas, e.g., Eq. (3), (4). In Lines 272-282, there is no Eq number.
3. The evaluation data is limited. The results are evaluated on only two benchmarks.
4. Figures 3 and 4 are not clear.
5. The proposed method is not novel. It seems to combine the search algorithm and RL.
6. The baselines are limited. Please compare with both open-source and closed-source SOTA LLMs, e.g., gpt5, Claude Sonnet 4.5, or Qwen, DeepSeek, etc.
7. In Table 2, the improvement of +reward is not significant.
8. The source code and data are missing.

**Questions:**

See Weaknesses.

---

### Note · Authors · 2026-01-13

I have read and agree with the venue's withdrawal policy on behalf of myself and my co-authors.